# The Effect of Epidural Analgesia on Labour and Neonatal and Maternal Outcomes in 1, 2a, 3, and 4a Robson’s Classes: A Propensity Score-Matched Analysis

**DOI:** 10.3390/jcm11206124

**Published:** 2022-10-18

**Authors:** Bruno Antonio Zanfini, Stefano Catarci, Francesco Vassalli, Valentina Laurita Longo, Matteo Biancone, Brigida Carducci, Luciano Frassanito, Antonio Lanzone, Gaetano Draisci

**Affiliations:** 1Department of Emergency, Anesthesiological and Reanimation Sciences, Fondazione Policlinico Universitario A Gemelli IRCCS, Largo A Gemelli 8, 00168 Roma, Italy; 2Obstetric Anesthesia, Department of Critical Care and Perinatal Medicine, Istituto di Ricovero e Cura a Carattere Scientifico (IRCCS), Istituto Giannina Gaslini, 16147 Genova, Italy; 3Department of Women, Children and Public Health Sciences, Fondazione Policlinico Universitario A Gemelli IRCCS, Largo A Gemelli 8, 00168 Roma, Italy; 4Università Cattolica del Sacro Cuore Roma, Largo F. Vito 1, 00168 Roma, Italy

**Keywords:** epidural analgesia, RTGCS, labour duration, second stage, Apgar scores, Caesarean section, operative vaginal delivery, propensity score-matched analysis

## Abstract

Background: Lumbar epidural analgesia (EA) is the most commonly used method for reducing labour pain, but its impact on the duration of the second stage of labour and on neonatal and maternal outcomes remains a matter of debate. Our aim was to examine whether EA affected the course and the outcomes of labour among patients divided according to the Robson-10 group classification system. Methods: Patients of Robson’s classes 1, 2a, 3, and 4a were divided into either the EA group or the non-epidural analgesia (NEA) group. A propensity score-matching analysis was performed to balance the intergroup differences. The primary goal was to analyse the duration of the second stage of labour. The secondary goals were to evaluate neonatal and maternal outcomes. Results: In total, 21,808 cases were analysed. The second stage of labour for all groups was prolonged using EA (*p* < 0.05) without statistically significant differences in neonatal outcomes. EA resulted in a lower rate of episiotomies in nulliparous patients, with a higher rate of operative vaginal deliveries (OVD) (*p* < 0.05) and Caesarean sections (CS) (*p* < 0.05) in some classes. Conclusions: EA prolonged the duration of labour without affecting neonatal outcomes and reduced the rate of episiotomies, but also increased the rate of OVDs.

## 1. Introduction

Lumbar epidural analgesia (EA) is the recognised gold standard in labour pain control [1]. However, with data available that both support and refute a relationship between EA and a significant prolongation of the second stage of labour [1,2,3,4] (especially with low-dose anaesthetic protocols), its role is still controversial [5,6]. The second stage of labour is described as the period between complete cervical dilatation and the delivery of the baby. In 2014, the American College of Obstetricians and Gynecologists (ACOG) defined the normal duration of the second stage of labour as up to 2 h in multiparous women and 3 h in nulliparous ones [7]. However, as long as progress is being documented [7,8,9], newer recommendations propose longer durations based on individual factors [10,11] such as parity, maternal age [2] and body mass index (BMI) [12], hypertension [13], foetal weight and position [14], maternal position [15], oxytocin augmentation [2], and EA [3]. This study’s focus on this phase of labour was strictly related to the potential impact of EA on foetal and maternal outcomes, and obstetric decision-making [16,17,18]. In the literature, some studies have reported no detrimental foetal outcomes in cases involving a longer duration [3,19,20], while others show increased rates of maternal morbidity (third- or fourth-degree perineal lacerations, postpartum haemorrhage, and chorioamnionitis) [16,21] and Caesarean sections (CS), with labour dystocia as one of the leading indications [18]. In order to investigate the effects of EA on labour effectively, a randomised controlled trial (RCT) involving women categorised according to their obstetric history would be ideal. However, as the randomisation required for an RCT presents ethical difficulties when working with healthy pregnant women, propensity score-matching is more appropriate. Propensity score-matching is a statistical method for collecting data retrospectively that minimises the selective biases that can arise from patients’ backgrounds. Many studies have reported that propensity score-matching produces similar results to RCTs despite its retrospective nature [4]. Based on five easily definable maternal characteristics, the Robson-10 classification system (RTGCS), introduced in 2001 and recognised by the WHO as the global standard for the analysis of pregnant patients [22], minimises bias by comparing pregnant populations both within and across institutions [23].

Using the propensity score-matching method, the present study aimed to analyse the impact of EA on the length of the second stage of labour and on foetal and maternal outcomes in the population of pregnant women referred to an Italian university hospital. These women have been stratified according to the RTGCS in order to settle the maternal characteristic confounders.

## 2. Methods

### 2.1. Study Population

A retrospective cohort analysis was performed of all live, at-term (37–42 weeks) vaginal deliveries at a tertiary university hospital over an 11-year period (October 2008 to October 2019). This population was divided according to the RTGCS. Research approval for this retrospective analysis was obtained from an institutional ethics committee without ad hoc consent from enrolled patients (Protocol ID 3741). The study was registered at ClinicalTrials.Gov (NCT 05579808). Pregnant patients were enrolled according to the following RTGCS groups: R1 (nulliparous, single, cephalic full-term pregnancy with spontaneous labour), R2a (nulliparous, single, cephalic full-term pregnancy with induced labour), R3 (multiparous, single, cephalic full-term pregnancy with spontaneous labour), and R4a (multiparous, single, cephalic full-term pregnancy with induced labour). The exclusion criteria for this study were cases involving multiple pregnancies, known major foetal or chromosomal abnormalities, pre-labour Caesarean deliveries, and elective Caesarean deliveries. The primary aim of this study was to evaluate the length of the second stage of labour, defined as the time from the first documented full dilatation to delivery, for pregnant women stratified according to the groups described above and either with epidural analgesia (EA) or without EA (non-epidural analgesia (NEA)). The secondary aims were to explore the potential associations between the duration of the second stage of labour and the length of first-stage labour, delivery modality, foetal outcomes (measured as resuscitation rates and Apgar score at 1 and 5 min), and maternal outcomes (measured as uterine atony, third- and fourth-degree lacerations, episiotomy, Caesarean section, and operative vaginal delivery rates) in the same groups.

### 2.2. Data Collection

All maternal and obstetrical data were prospectively collected by labour and delivery unit personnel by entering cases into a perinatal database, which were then cross-tabulated on an Excel file. The collected data included demographic and obstetric parameters: maternal age and BMI, hypertension, diabetes, foetal weight and position, gestational age, labour induction (intravaginal or intracervical prostaglandin E2 gel, oxytocin), operative vaginal delivery (OVD) (only via the Kiwi OmniCup (produced by Clinical Innovations, Muray, UT, USA) vacuum extractor), Caesarean section (CS), maternal morbidity (uterine atony, episiotomy, third- to fourth-degree perineal laceration), and foetal morbidity (Apgar score <7 at 1 and 5 min, neonatal resuscitation).

### 2.3. Epidural Analgesia Method

Access to EA is available on a 24-h basis, with protocols reserving its administration for consenting women previously informed in an epidural outpatient clinic. All women who request analgesia for pain relief during labour are evaluated by an anaesthetist for suitability. Patients meeting absolute (i.e., uncorrected hypovolemia, coagulopathy, anticoagulant therapy) or relative (i.e., anatomical deformities, certain neurological disorders, sepsis) exclusion criteria are not qualified to receive EA. During labour, in the presence of a cervical dilatation of ≥3 cm and in the active phase of the first stage (established by a partograph), maternal status (blood pressure and temperature) and foetal well-being (20 min of normal cardiotocography) are also evaluated. In the absence of abnormalities, intravenous access by a 14-gauge (G) or 16G cannula is positioned and a crystalloid infusion is started. Using an aseptic technique while the patient is in a sitting position, an epidural catheter is then placed at the L2–L3 or L3–L4 interspace. Finally, analgesia is established with the epidural administration of a low dose of local anaesthetic, plus a lipid-soluble opioid (ropivacaine 0.1% and sufentanil 0.5%, 20 mL). Maternal blood pressure, foetal heart rate, pain scores, and the extent of sensory block are then assessed at five-minute intervals for the first 15 min, then every half-hour. Analgesia is maintained with a top-up regimen, using intermittent manual epidural boluses of increasing concentrations of ropivacaine, with up to 0.15% at full dilation, according to the specific needs of individual participants.

### 2.4. Statistical Analysis

As the use of observational data prevents the control of treatment assignment through randomisation, systematic differences may be present between treated and untreated subjects in an RCT. This can introduce confounding and thus prevent associations from being reliably estimated. However, the propensity score-matching method is a statistical technique that aims to predict the probability of receiving treatment and/or having a condition based on identified covariates and background characteristics. Thus, the study population was matched using a propensity score model with the optimal algorithm, including maternal age, BMI, gestational age, neonatal weight, parity, diabetes, and hypertension. The algorithm excluded patients with missing data in the matching variables. The quality of the matching was verified by considering an acceptable standard mean difference of <0.1 that was obtained in all participant groups. Data are reported by mean ± standard deviation or by rate (i.e., percentage), unless indicated otherwise. All numerical variables with normal distributions were compared via t-tests, while categorical variables were compared via the chi-squared test or Fisher’s test when the expected frequencies were less than 5. Statistical significance was considered for *p*-values < 0.05. All analyses were performed with R statistical computing software (R Foundation for Statistical Computing, Austria, 4.1.2).

## 3. Results

During the study period, 30,739 pregnant women were admitted to the delivery room. Among them, 8931 were excluded due to the exclusion criteria or an incomplete data record. The remaining 21,808 cases were divided according to Robson’s groups: R1 (8,359 patients), R2a (4,739 patients), R3 (6,805 patients), and R4a (1,905 patients). The distribution of EA among nulliparous (groups R1 and R2a, 63%) and multiparous (groups R3 38% and R4a, 42%) women is presented in Figure 1.

Optimal propensity score-matching rendered a total of 2210 pairs in the R1 group, 1294 pairs in R2a, 1934 pairs in R3, and 639 pairs in R4a. The demographic characteristics of the participants are presented in Table 1.

### 3.1. Labour Length

A statistically significant prolongation of the duration of the second stage of labour was reported among patients receiving EA in all four groups (Table 2). The mean second-stage lengths reported were as follows. In the R1 group, it was 77 ± 59 min in EA patients and 54 ± 44 min in NEA patients (*p* < 0.01); in the R2a group, it was 85 ± 63 min in EA patients and 55 ± 49 min in NEA patients (*p* < 0.01); in the R3 group, it was 32 ± 30 min in EA patients and 22 ± 18 min in NEA patients (*p* < 0.01); and in the R4a group, it was 34 ± 35 min in EA patients and 20 ± 21 min in NEA patients (*p* < 0.01). The active phase of first-stage labour was also significantly prolonged, with the mean values reported in Table 2.

### 3.2. Neonatal and Maternal Outcomes

The prolonged length of the second stage of labour did not result in any adverse neonatal outcomes. Differences in resuscitation rates and Apgar scores at 1 min and 5 min were not statistically significant between the EA and NEA groups, and the number of Apgar < 7 scores at 5 min were reduced in the R2 epidural group (EA) (Table 3).

Regarding maternal outcomes, EA was protective against episiotomy in nulliparous patients (groups R1 and R2a, *p* < 0.05). Differences in the rates of uterine atony or third- to fourth-degree laceration between the EA and NEA groups were not statistically significant across the whole study population. A statistically significant increase in OVD was reported among patients receiving EA in the R1, R2a, and R3 groups (*p* < 0.05). A statistically significant increase in CS rates among pregnant women receiving EA in the R2a and R3 groups was also reported (Table 4).

## 4. Discussion

The results of this study showed the following.

-EA prolongs the second stage of labour in nulliparous and multiparous women with and without labour induction.-A longer labour length is not correlated with any adverse neonatal outcomes.-Increases in OVDs are statistically significant among patients who received EA in all four groups.-There is an increased risk of CS for R2a and R3 patients with EA.-Episiotomy rates are reduced among nulliparous women with EA, with no statistically significant difference in atony or pelvic laceration rates.

To our knowledge, a small number of published studies have investigated the effect of EA on labour using propensity score-matching [4], and none has analysed such data according to Robson’s classes. This study has found an increase in the length of the second stage of labour among patients receiving EA, as reported in other studies. Nowadays, a prolonged second-stage length is accepted as normal for patients who have received EA, and new partographs allow an additional hour for this subgroup. Recent studies even propose to allow any duration, as long as there is a safe progression of labour [8]. The results of this study confirm this trend. The mechanism of prolongation in the second stage of labour among women receiving EA is not well defined; however, it is thought to include the direct effects of an epidural on uterine contractility by its suppression of PGF2α. This effect is consistent with a significant reduction in the frequency of contractions recorded by EMG activity, compared between women with and without EA [24]. Additionally, the lack of the spontaneous urge to push among women with EA may cause a delay in active pushing [2,17]. Though a prolonged second stage of labour was found by this study, detrimental neonatal outcomes were not reported. This underscores the safety of the epidural technique for labour analgesia. Published data on the impact of EA for labour pain relief on neonatal outcomes are contradictory. A recent Cochrane review of RCTs on the effectiveness and safety of EA showed that EA did not appear to have an immediate effect on neonatal status as determined by Apgar scores < 7 at five minutes [1]. Similar results in relation to the Apgar score have been reported by Wang in a Chinese academic medical facility [25]; a non-significant influence of EA on neonatal well-being in terms of neuroendocrine response to stress was also reported in [26]. However, a Dutch propensity analysis found an association among EA, low Apgar scores, and more neonatal intensive care admissions [20]. The latter report suggests that opioids can diffuse from the epidural space into the maternal blood and placenta, which can affect the respiratory centre of the neonate [27]. According to other recently published studies [28], adding opioids to the local anaesthetic effectively enhances the analgesic effect and thus reduces the consumption of local anaesthetic. No influence on neonatal outcomes was reported in the present analysis. Conversely, OVDs were increased among patients with EA (with statistical significance in the R1, R2a, and R3 groups), in accordance with many published studies. The main reason for this mode of delivery could be linked to the increased length of labour [29]. Finally, this study found an increase in Caesarean rates in the R2a and R3 groups, in contrast to data in the literature [4,30]. This is a controversial finding that should be further analysed. It could be speculated that even when accepting the relationship between EA and prolonged labour, other obstetric factors such as changes in the physicians’ behaviour and non-medical risk factors could interfere with the final decision-making process. Even though epidural use has been associated with a prolonged second stage of labour and a higher rate of OVD, both being known risk factors for vaginal lacerations [31], an increase in perineal tears in pregnant women receiving EA was not found. This is also in agreement with authors who have suggested protective effects related to EA as result of reduced pain and a more controlled delivery [32]. The protective effects of EA were also reported in relation to episiotomy rates, which were reduced in nulliparous women who had received an epidural. This is an important outcome, as this technique does not prevent pelvic floor damage [33,34] and is associated with dyspareunia and anal incontinence.

Even though this study has benefited from a sound statistical analysis and was performed rigorously by using the RTGCS to minimise bias among pregnant patients, there are some important limitations to consider when interpreting its results. Firstly, while the analyses were performed after propensity score-matching to reduce intergroup differences, the results may have still been affected by confounders that were not collected. These included important outcomes, such as measurements of anaesthetic levels during EA and subjective pain scores throughout labour. Secondly, although the majority of obstetricians adhered to the guidelines that strictly define the threshold for assisted vaginal delivery and Caesarean section, some differences among the individual caregivers engaged in clinical practice over the long study period occurred. Further clinical trials are needed to confirm our results.

## 5. Conclusions

The findings in the present retrospective analysis indicate that EA prolongs the second stage of labour in both nulliparous and multiparous women with or without labour induction, without affecting neonatal outcomes. A significant increase in OVDs, a significant decrease in episiotomy rates in nulliparous women, and a significant increase in Caesarean section rate in some Robson’s classes were also reported in pregnant women using EA.

## Figures and Tables

**Figure 1 jcm-11-06124-f001:**
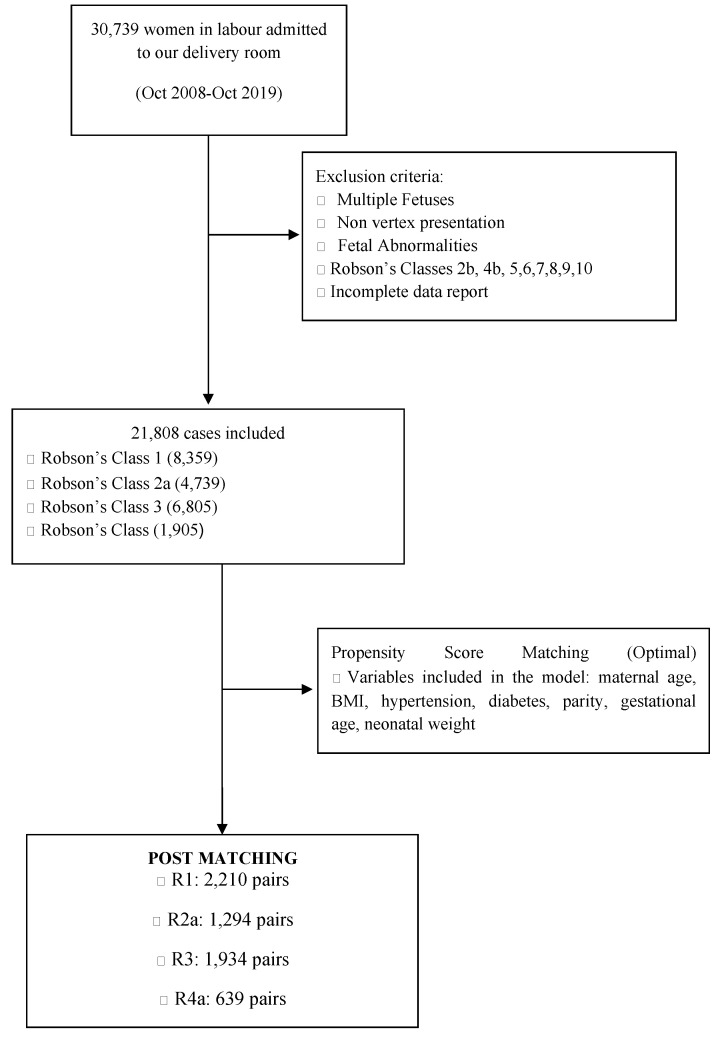
Flow chart of enrolment.

**Table 1 jcm-11-06124-t001:** Study population characteristics in Robson’s classes 1, 2a, 3, and 4a. Data are expressed as the mean ± standard deviation or cases (percentage). EA: epidural analgesia group; NEA: non-epidural analgesia group.

	Robson 1		Robson 2a		Robson 3		Robson 4a	
	EA (n = 5226)	NEA (n = 3133)	EA (n= 2996)	NEA (n= 1743)	EA (n = 2612)	NEA (n = 4193)	EA (n = 791)	NEA (n = 1114)
Maternal age (years)	32 ± 5	30 ± 6	33 ± 5	31 ± 5	34 ± 4	33 ± 5	35 ± 5	34 ± 5
BMI (kg/m^2^)	22 ± 3.6	22.3 ± 4.1	22.9 ± 4.3	23.4 ± 4.9	22.6 ± 3.6	23 ± 4.1	23.8 ± 5.2	24.1 ± 5.1
Parity (n)	0	0	0	0	1.2 ± 0.5	1.4 ± 0.9	1.2 ± 0.5	1.3 ± 0.7
Gravidity (n)	1.3 ± 0.6	1.3 ± 0.7	1.3 ± 0.7	1.3 ± 0.7	2.6 ± 1	2.9 ± 1.3	2.7 ± 1	2.9 ± 1.3
Foetal weight (g)	3287 ± 380	3227 ± 416	3284 ± 423	3209 ± 451	3410 ± 407	3370 ± 429	3385 ± 446	3315 ± 464
Gestational age (week)	40.3 ± 1.1	40.1 ± 1.1	40.1 ± 1.2	40.1 ± 1.2	40.2 ± 1	40.1 ± 1.1	40.1 ± 1.2	40 ± 1.2
Hypertension	95 (1.8)	69 (2.2)	183 (6.1)	125 (7.2)	30 (1.1)	73 (1.7)	55 (7)	90 (8.1)
Diabetes	260 (5)	186 (5.9)	429 (14.3)	297 (17)	111 (4.2)	252 (6)	152 (19.2)	213 (19.1)

BMI, body mass index; R1, nulliparous, single, cephalic full-term pregnancy with spontaneous labour; R2a, nulliparous, single, cephalic full-term pregnancy with induced labour; R3, multiparous, single, cephalic full-term pregnancy with spontaneous labour; R4a, multiparous, single, cephalic full-term pregnancy with induced labour.

**Table 2 jcm-11-06124-t002:** Length of labour in Robson’s classes 1, 2a, 3, and 4a after propensity score-matching.

	Robson 1 (2210 Pairs)			Robson 2a (1294 Pairs)			Robson 3 (1934 Pairs)			Robson 4a (639 Pairs)		
	EA	NEA	*p*-Value	EA	NEA	*p*-Value	EA	NEA	*p*-Value	EA	NEA	*p*-Value
Stage I, active phase (min)	213 ± 143	137 ± 128	<0.001	205 ± 154	141 ± 141	<0.001	129 ± 90	84 ± 83	<0.001	120 ± 95	87 ± 86	<0.001
Stage II (min)	77 ± 59	55 ± 49	<0.001	85 ± 63	55 ± 49	<0.001	32 ± 30	22 ± 18	<0.001	34 ± 35	20 ± 21	<0.001

R1, nulliparous, single, cephalic full-term pregnancy with spontaneous labour; R2a, nulliparous, single, cephalic full-term pregnancy with induced labour; R3, multiparous, single, cephalic full-term pregnancy with spontaneous labour; R4a, multiparous, single, cephalic full-term pregnancy with induced labour.

**Table 3 jcm-11-06124-t003:** Neonatal outcomes in Robson’s classes 1, 2a, 3, and 4a after propensity score-matching. The *p*-values refer to the chi square or Fisher’s test (#).

	Robson 1 (n= 2210 Pairs)			Robson 2a (n = 1294)			Robson 3 (n = 1934)			Robson 4a (n = 639)		
	EA	NEA	*p*-Value	EA	NEA	*p*-Value	EA	NEA	*p*-Value	EA	NEA	*p*-Value
Apgar 1 min < 7	52 (2.4)	34 (1.5)	0.064	27 (2.1)	37 (2.9)	0.255	20 (1)	24 (1.2)	0.649	8 (1.3)	8 (1.3)	>0.999
Apgar 5 min < 7	4 (0.2)	7 (0.3)	0.548 #	1 (0.1)	9 (0.7)	0.021 *	3 (0.2)	2 (0.1)	>0.999 #	0 (0)	5 (0.8)	0.062 #
Resuscitation	53 (2.4)	53 (2.4)	>0.999	31 (2.4)	27 (2.1)	0.69	22 (1.1)	34 (1.8)	0.139	10 (1.6)	13 (2)	0.674

R1, nulliparous, single, cephalic full-term pregnancy with spontaneous labour; R2a, nulliparous, single, cephalic full-term pregnancy with induced labour; R3, multiparous, single, cephalic full-term pregnancy with spontaneous labour; R4a, multiparous, single, cephalic full-term pregnancy with induced labour. *: statistically significant.

**Table 4 jcm-11-06124-t004:** Maternal outcomes in Robson’s classes 1, 2a, 3, and 4a after propensity score-matching. The *p*-values refer to chi square or Fisher’s tests (#).

	Robson 1 (n = 2210 Pairs)			Robson 2a (n = 1294)			Robson 3 (n = 1934)			Robson 4a (n = 639)		
	**EA**	**NEA**	** *p* ** **-Value**	**EA**	**NEA**	** *p* ** **-Value**	**EA**	**NEA**	***p*-Value**	**EA**	**NEA**	***p*-Value**
Uterine atony	121 (5.5)	142 (6.4)	0.203	102 (7.9)	117 (9)	0.323	78 (4)	98 (5.1)	0.143	47 (7.4)	49 (7.6)	0.915
III–IV laceration	11 (0.5)	11 (0.5)	1	6 (0.5)	11 (0.9)	0.33	3 (0.2)	4 (0.2)	1 #	2 (0.3)	2 (0.3)	1 #
Episiotomy	1227 (55.5)	1352 (61.2)	<0.001	655 (50.6)	736 (56.9)	0.002	753 (38.9)	806 (41.7)	0.088	221 (0.3)	228 (35.7)	0.725
Caesarean section	47 (2.1)	42 (1.9)	0.668	46 (3.6)	28 (2.2)	0.045	12 (0.6)	2 (0.1)	0.013 #	8 (1.3)	3 (0.5)	0.225 #
Operative vaginal delivery	293 (13.3)	151 (6.8)	<0.001	201 (15.5)	117 (9)	<0.001	33 (1.7)	30 (1.6)	0.783	18 (2.8)	11 (1.7)	0.251

R1, nulliparous, single, cephalic full-term pregnancy with spontaneous labour; R2a, nulliparous, single, cephalic full-term pregnancy with induced labour; R3, multiparous, single, cephalic full-term pregnancy with spontaneous labour; R4a, multiparous, single, cephalic full-term pregnancy with induced labour.

## Data Availability

Data presented in this study are available on request from the corresponding author.

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
