# Peer review of "The Effect of Epidural Analgesia on Labour and Neonatal and Maternal Outcomes in 1, 2a, 3, and 4a Robson’s Classes: A Propensity Score-Matched Analysis"

_jcm, 2022, doi:10.3390/jcm11206124_

Round 1
Reviewer 1 Report
the manuscript is interesting and well written. I would include in the summary of results the prolongation of stages of labor in minutes, number of episiotomies and c sections.
Author Response
Reviewer 1
the manuscript is interesting and well written. I would include in the summary of results the prolongation of stages of labor in minutes, number of episiotomies and c sections.
Thank you for your comments. Due to limited number of words allowed for abstract (200 words maximum), unfortunately we have been not able to include in the summary of results what suggested.
Reviewer 2 Report
I read with great interest the manuscript by Zanfini et al. on the effect of epidural analgesia on labour and neonatal and maternal outcomes in 1, 2a, 3, and 4a Robson’s classes. The article is interesting and well written. However, I have some minor issues to be addressed by the authors:
Minor issues
- In the methods section (paragraph “study population” it is not clear which “foetal and maternal outcomes” the authors take into account. Please specify.
- Table 2, 3 and 4 are hard to read. Please modify.
- Two recent studies (Wang et al. 2018 DOI: 10.1007/s00404-018-4777-6 and La Camera et al. 2021 https://doi.org/10.1080/01443615.2020.1755621) provide consistent results to the ones reported in this paper regarding the effect of EA on neonatal outcomes. Please add them to the discussion section.
- Please add a conclusion section at the end of the paper.
- The first two names in reference number 12 are misspelled. Please fix it.
Author Response
Reviewer 2
I read with great interest the manuscript by Zanfini et al. on the effect of epidural analgesia on labour and neonatal and maternal outcomes in 1, 2a, 3, and 4a Robson’s classes. The article is interesting and well written. However, I have some minor issues to be addressed by the authors:
Minor issues
- In the methods section (paragraph “study population” it is not clear which “foetal and maternal outcomes” the authors take into account. Please specify.
Thank you for your suggestion. We modified the text better specifying as foetal outcomes the resuscitation rates and Apgar’s score at 1 and 5 minutes; as maternal outcomes uterine atony, 3rd and 4th degree lacerations, episiotomy, cesarean section and operative vaginal delivery rates.
- Table 2, 3 and 4 are hard to read. Please modify.
Thank you. We slightly modified the tables to make them easy to read.
- Two recent studies (Wang et al. 2018 DOI: 10.1007/s00404-018-4777-6 and La Camera et al. 2021 https://doi.org/10.1080/01443615.2020.1755621) provide consistent results to the ones reported in this paper regarding the effect of EA on neonatal outcomes. Please add them to the discussion section.
According to your suggestion, we modified the Discussion section adding the results of suggested papers.
- Please add a conclusion section at the end of the paper.
Thank you for your suggestion. We added a conclusion section at the end of the paper.
- The first two names in reference number 12 are misspelled. Please fix it.
Thank you. We fixed the names in reference 12.
Reviewer 3 Report
Dear Authors,
Congratulations on a very well-done piece of scientific job. Nevertheless, I shall some comments:
1. I wonder if it were o good idea to register the study in Clinical Trial Registy, even retrospectively.
2. in the methodology I would appreciate to read a more detailed information concerning the dosage of epidural analgesia. I doubt if all pregant patients received 20 ml of anaesthetic mixture, despite their height. I would more precisely state if Bromage rule or to-up regimen was applied, according to specific needs of individual participants.
3. I would explain Robson classification under table 2. I know full well that it is obvious for practitioners. Nevertheless, as an anaesthesiologist, I must explain Apfel score of PONV and ASA score in every publication, what I personally find irritating, but some readers may appreciate it.
Looking forward to your fast corrections, because I am planning a study with influence of epidural anaestesia on the perioperative outcomes and would be pleased to cite your well-done research.
I would improve the flow chart of the study protocol.
Best regards
MS
Author Response
Reviewer 3
Dear Authors,
Congratulations on a very well-done piece of scientific job. Nevertheless, I shall some comments:
- I wonder if it were o good idea to register the study in Clinical Trial Registy, even retrospectively.
Thank you for your suggestion. We registered our study in Clinical Trial Registry, as specified in the paper.
- in the methodology I would appreciate to read a more detailed information concerning the dosage of epidural analgesia. I doubt if all pregant patients received 20 ml of anaesthetic mixture, despite their height. I would more precisely state if Bromage rule or to-up regimen was applied, according to specific needs of individual participants.
Thank you. We modified the text specifying that a top-up regimen was used.
- I would explain Robson classification under table 2. I know full well that it is obvious for practitioners. Nevertheless, as an anaesthesiologist, I must explain Apfel score of PONV and ASA score in every publication, what I personally find irritating, but some readers may appreciate it.
As suggested, we added Robson’s classification under Tables, to make them easy to read.